# Modelling the Influence of Slide Burnishing Parameters on the Surface Roughness of Shafts Made of 42CrMo4 Heat-Treatable Steel

**DOI:** 10.3390/ma14051175

**Published:** 2021-03-02

**Authors:** Rafał Kluz, Katarzyna Antosz, Tomasz Trzepieciński, Magdalena Bucior

**Affiliations:** 1Department of Manufacturing and Production Engineering, Rzeszow University of Technology, al. Powst. Warszawy 8, 35-959 Rzeszów, Poland; rkluz@prz.edu.pl (R.K.); magdabucior@prz.edu.pl (M.B.); 2Department of Materials Forming and Processing, Rzeszow University of Technology, al. Powst. Warszawy 8, 35-959 Rzeszów, Poland; tomtrz@prz.edu.pl

**Keywords:** average surface roughness, plastic working, slide burnishing, steel shaft, surface topography

## Abstract

This article presents the results of tests aimed at determining the effect of slide burnishing parameters on the surface roughness of shafts made of 42CrMo4 heat-treatable steel. The burnishing process was carried out using tools with polycrystalline diamond and cemented carbide tips. Before burnishing, the samples were turned on a turning lathe to produce samples with an average surface roughness *Ra* = 2.6 µm. The investigations were carried out according to three-leveled Hartley’s poly selective quasi D (PS/DS-P: Ha3) plan, which enables a regression equation in the form of a second-order polynomial to be defined. Artificial neural network models were also used to predict the roughness of the surface of the shafts after slide burnishing. The input parameters of the process that were taken into account included the values of pressure, burnishing speed and feed rate. Overall, the burnishing process examined leads to a reduction in the value of the surface roughness described by the *Ra* parameter. The artificial neural networks with the best regression statistics predicted an average surface roughness of the shafts with *R*^2^ = 0.987. The lowest root-mean-square error and mean absolute error were obtained with all the network structures analysed that were trained with the quasi Newton algorithm.

## 1. Introduction

Slide burnishing (SB) is a finishing method that consists in local plastic deformation of the material as a result of static or kinetic interaction of the tool surface with the workpiece surface. Burnishing treatment provides the required dimensions and shape of the elements, but it is not very advantageous from the point of view of ensuring satisfactory the stereometric surface structure in the final stage of the production of the element [1]. Due to the small dimensions of the tool tips, the plastic deformation zone of the burnished parts is very small. Depending on the purpose of the processing, slide burnishing can be divided into hardening, smoothing or dimensional burnishing [2]. SB is a method of precision machining that is used to improve mechanical properties, corrosion resistance, surface roughness, wear resistance and fatigue strength [3]. Slide burnishing creates a deep hardened layer in which a state of compressive stresses is generated [4]. Due to the small radius of the tool, SB is characterized by weak forces, which permits the machining of parts with low rigidity [5]. The burnishing process allows the following benefits to be achieved [6]: increase in surface hardness, increase in fatigue resistance, possibility of processing surfaces with a large radius corner, possibility of processing surfaces with a low coefficient of friction, ability to produce a high degree of smoothness on a surface, minimal heating of the surface layer during processing, possibility of using burnishing tools mounted on universal lathes, high process efficiency and low energy consumption of the SB process.

In recent years, many authors have carried out studies on the influence of the parameters of the burnishing process on the properties of the surface layer using various research techniques. Korzyński et al. [7] carried out sliding burnishing on 42CrMo4 alloy steel with a cylindrical sliding element made of polycrystalline diamond. After treatment, the surface layer was shown to have increased microhardness and internal compressive stresses. The influence of the input parameters of the SB on the surface roughness and residual stresses of the 42CrMo4 steel surface was also investigated by Dzierwa and Markopoulos [8]. The tests confirmed that the value of the root mean square roughness parameter of the treated surface decreased from 0.522 to 0.051 μm. The wear resistance of the treated surface has also been improved. The research conducted by Świrad [9] proved that the roughness of the surface of workpieces made of 42CrMo4 steel can be significantly reduced thanks to an SB process carried out with a Ti3SiC2-based composite polycrystalline diamond tool. Gharbi et al. [10] applied the Taguchi method to evaluate and optimize the influence of burnishing parameters on the roughness and hardness of surfaces subjected to SB. The mean roughness of AISI 1010 hot rolled sheets was improved from *Ra* = 2.48 to 1.75 μm, while the hardness increased from 59 to 65.5 HRB. Luo et al. [11] analysed SB of copper alloys using a polycrystalline diamond sliding tool. The results showed that a low value of the roughness parameter does not mean a low surface roughness or waviness, and that optimal burnishing parameters can be obtained with various combinations of processing parameters. Shiou and Cheng [12] used the Taguchi L18 orthogonal plan to analyse the ball burnishing process of a NAK80 tool steel mould surface. Due to the use of optimal burnishing parameters, the mean roughness *Ra* of the part surfaces decreased from about 1.0 to 0.020 μm. Maximov et al. [13] analysed the influence of the parameters of the slide burnishing process of AISI316Ti steel on the value of residual stresses and wear resistance of the treated surfaces. The higher burnishing speed increased the burnishing productivity, but on the other hand this speed decreased the value of the residual stresses. El-Tayeb et al. [14] applied the burnishing process to the surface treatment of shafts made of 6061 aluminium alloy. It was found that the appropriate burnishing speed can improve surface roughness by up to 40%. Burnishing allowed for an almost 46% reduction in the coefficient of friction in relation to the untreated surface. Silva-Alvarez et al. [15] proposed the use of the ball burnishing process in order to improve the surface properties of a cobalt-chromium-molybdenum (CoCrMo) alloy. A 32-factor experimental design was carried out to understand the relationship between surface hardness, surface roughness and impact force and the number of tool passes. Statistical analysis showed that the burnishing force is the most important factor changing the surface properties. After applying the optimal processing parameters hardness increased by 41.4%, and roughness decreased by 72.7%. A comprehensive literature review of publications on slide burnishing is presented by Maximov et al. [16].

Recently, with the development of burnishing technology, Low Plasticity Burnishing (LPB^®^) has been proposed. LPB^®^ is Lambda Technologies Group’s (Cincinnati, Ohio, OH, USA) most popular method of introducing designed compressive residual stresses to improve component performance [17]. LPB imparts a designed residual stress field into the surface of components to produce a stable layer of beneficial residual compression. In LPB processes, the most distinctive feature is that a ceramic ball or a carbide alloy is stimulated by the high-pressure liquid. As a result, a deeper residual compressive stress and better *Ra* were obtained [18]. The LPB process can efficiently improve the micro-hardness of the surface, corrosion resistance, ultimate tensile strength, surface finish, low- and high-cycle fatigue strength and wear resistance [19,20]. A comprehensive review of the LPB, ball burnishing and roller burnishing processes and the work carried out by various researchers in these areas has been provided by Priyadarsini et al. [21]. Prevéy and Jayaraman [22] presented a brief overview of the improved damage tolerance achievable with LPB for various materials and damage mechanisms.

Ball burnishing can be applied to improve the mechanical and surface properties after the friction stir welding (FSW) process [23]. This technique is a potential solution that can enhance the surface properties by deforming the welding area that commonly presents different thermomechanically affected zones [24]. In particular, Prevey et al. [25] investigated the fatigue improvement of aluminium alloys first welded by FSW and later low plasticity burnished as a final surface treatment. It was found that the burnishing process induces residual compression stresses which tend to improve the fatigue behaviour of the component. López de Lacalle [26] found that a higher pressure in the burnishing process promotes the appearance of grooves on the surface of Inconel 718 due to the ductility of this alloy. In another study [27], they found that ball burnishing is an effective surface treatment in reducing the time of finishing the operation. Ball burnishing is considered to be an interesting technology to reinforce the strength of the welding area that has suffered a thermal softening due to the FSW process [24].

Machine shafts, including those made of 42CrMo4 steel, work in heavy-duty conditions and with high loads and rotational speeds. The quality and duration of their work largely depend on the technology used in their production. The exploitation properties, including abrasion resistance, fatigue life and corrosion resistance depend on the accuracy and properties of the surface layer. Slide burnishing has many advantages, such as securing increased hardness, corrosion resistance, and fatigue life as a result of producing compressive residual stresses in surface layer of shafts made of heat-treatable alloy steel, i.e., 42CrMo4. Surface roughness affects the fatigue life of burnished surfaces in a great extent. Due to the fact that the surfaces after burnishing have several times greater radii of asperities rounding, the corrosion resistance increases by about 20% [28]. Appropriate selection of the parameters of the burnishing process related to the mutual interactions of many parameters is the most important aspect in assessing good performance in service. Thus, controlling of the burnishing conditions in such a way as to produce roughness that is as smallest as surface could lead to considerable improvement in component life.

ANN (Artificial neural networks) are computing systems for the analysis of linear and non-linear complex regression problems, as well as for real-time supervision of machining processes [29]. Artificial neural networks (ANN) have been widely employed for predicting the behaviour of distinct processes with regard to cutting forces, residual stresses, as well as machined surface roughness. There are many ANN structures used to model the relationship between input and output parameters [30,31]. The neural network consists of at least two layers, input and output, and an arbitrary number of intermediate layers, called hidden layers [32]. The number of neurons in the input and output layers is determined by the number of input and output parameters. The majority of the investigations existing in the literature on the effect of burnishing parameters on the burnished surface has been experimented in nature [33]. Very few analitycal models are available in literature [34,35,36]. The use of artificial neural network is purely based on expert knowledge and does not depend on complicated analytical systems. Due to exitence of many links in their structure, ANNs are more effective compared to multiple regression analysis [37], Taguchi’s method [38] and multi-response optimization [39,40]. Based on abductive modeling techniques, the neural networks represent sophisticated and uncertain relationship between input and output variables. Accordingly, the network can solve complex nonlinear problems using historical and current data.

Investigations on the effect of SB parameters on the quality of the treated surfaces have been carried out using artificial neural networks (ANN) which, with the selection of the appropriate structure, are able to model any non-linear relationships between the input and output parameters. In recent years, several general studies have been conducted [41,42]. Therefore the process of ANN application to analyse the burnishing process still requires improvement. The influence of the SB process on the surface topography of 42CrMo4 steel shafts was investigated in this study. The burnishing process was performed with the use of two tool tip materials, i.e., polycrystalline diamond and cemented carbide. The investigations were carried out on the basis of Hartley’s PS/DS-P:Ha3 plan, which enables the definition of a regression equation in the form of a second-order polynomial. The multi-layer perceptron was also used to investigate the correlation between the processing parameters and the surface roughness of the specimens.

## 2. Material and Methods 

### 2.1. Material

The test materials were shafts with a diameter of 50 mm made of 42CrMo4 alloy steel (Cognor S.A., Stalowa Wola, Poland) with a hardness of 21–22 HRC. This 42CrMo4 heat-treatable steel is a commonly used high hardenability chromium molybdenum steel that is typically used after quenching and tempering. Examples of applications for this steel are crankshafts, heavily loaded bolts, gears, axles and discs. The chemical composition of the tested steel (42CrMo4) specified in [43] is: C (0.18–0.45), Mn (0.40–0.70), Si (0.17–0.37), P (max. 0.035), S (max. 0.035), Cr (0.9–1.2), Ni (max. 0.3), Mo (0.15–0.25), W (max. 0.2), V (max. 0.05), Cu (max. 0.25). The basic mechanical properties are presented in Table 1.

### 2.2. Methods

The shafts were turned in such a way as to obtain the surface roughness defined as the average surface roughness *Ra* = 2.6 µm. The slide burnishing was performed on an LZ 360 universal lathe (Zakłady Mechaniczne, Tarnów, Poland), which is adapted to the precise production of medium-sized parts in accordance with the accuracy specified in DIN8605. The SB process was carried out on a test stand (Figure 1a) using a DB-3 burnishing tool (Figure 1b, Cogsdill-Nuneaton Ltd, Nuneaton, England). This study used a tool tip made of polycrystalline diamond and cemented carbide.

The experimental investigations were carried out in accordance with Hartley’s PS/DS-P:Ha3 static plan. This is a three-level experimental design that requires the establishment of input factors on three equally spaced levels. The process input parameters considered include the impact force, burnishing speed and feed rate. The matrix plan of experiments is presented in Table 2. Surface roughness measurements of the samples were carried out using a Taylor-Hobson Surtronic 2 profilometer (Taylor Hobson Ltd, Leicester, England) according to [44]. The average surface roughness *Ra*, the main parameter of the surface roughness, was chosen to describe the tribological properties of the burnished surfaces. As a result of the experiments carried out in accordance with the Hartley’s plan, the regression equation obtained for the SB process was:(1)$$y=b_{0}+\sum b_{k}x_{k}+\sum b_{kk}x_{k}^{2}+\sum b_{kj}x_{k}x_{j}$$
where *b_k_*, *b_kk_*, *b_kj_* are the coefficients in the regression equation, *x_k_* and *x_j_* are the input variables.

### 2.3. Modelling Using Artificial Neural Networks

The Statistica program was used to model the effect of the burnishing process parameters on the value of the average surface roughness *Ra* using ANN. A multilayer network (multilayer perceptron) with an appropriate number of hidden layers and neurons in these layers is capable of analysing and predicting any non-linear function. The hyperbolic tangent function was used to calculate the output value of neurons. The network learning process was carried out using three algorithms: a commonly used multi-layer network learning algorithm—back propagation (BP), the Levenberg-Marquardt (LM) algorithm used to train unidirectional networks, and the quasi Newton (qN) method. In the investigations the back propagation algorithm with a value of the learning coefficient of 0.1 was used for ANN training and 20% of the data included in the training set was assigned to the verification set. As a result of the training process, the ANN acquires prediction of the output signal based on the sequence of input signals and the corresponding output signals. In these investigations, the quality of the network was assessed on the basis of three parameters [41,42,45]:
(a)root mean square error *RMSE*:
(2)$$RMSE=\sqrt{\frac{1}{n}\overset{n}{\underset{i\;=\;1}{\sum }}{\left|a_{j}-p_{j}\right|}^{2}}$$
(b)coefficient of determination *R*^2^:
(3)$$R^{2}=1-\left(\frac{\sum_{i\;=\;1}^{n\;}{\left(a_{j}-p_{j}\right)}^{2}}{\sum_{i\;=\;1}^{n\;}{\left(p_{j}\right)}^{2}}\right)$$
(c)mean absolute error *MAE*:
(4)$$MAE=\frac{1}{n}\sum_{i\;=\;1}^{n}\left|a_{j}-p_{j}\right|$$
where *p* is the predicted value, *a* is the actual value, and *n* is the number of training sets.

The input data was normalized using the *min − max* function, which transforms the raw (original) data values into a new interval (*N_min_*, *N_max_*) using a linear function:(5)$$D^{\prime }=\frac{\left(D-min\right)}{max-min}\left(N_{max}-N_{min}\right)+N_{min}$$
where (*min*, *max*) is the interval in which the original data are contained, *D*—value of the variable subjected to normalization.

## 3. Results and Discussion

### 3.1. Hartley’s PS/DS-P:Ha3 Plan

The process of slide burnishing with a diamond tool has led to a significant reduction in the average surface roughness *Ra* of the shafts (Table 3). The greatest reduction of the *Ra* parameter was found after burnishing with a feed rate of *f* = 0.032 mm/rev., impact force *P* = 130 N and a burnishing speed *v* = 180 rpm. The greatest surface roughness of the shaft was observed after sliding burnishing with *f* = 0.063 mm/rev., *P* = 30 N and *v* = 270 rpm (Table 3). It can be concluded that the best surface roughness was obtained in experiment no. 3, and the worst in experiment no. 2 (Table 4). However, in each SB experiment the roughness *Ra* decreased. The difference between the *Ra* parameter values measured on two surfaces of shafts subjected to burnishing with the two tool tip materials used was very similar. This confirms the high repeatability of the burnishing process. The assessment of the significance of the coefficients in the regression equation was performed using the Student’s *t*-test, comparing them with the calculated test values (Table 5 and Table 6).

The adequacy of the regression equation obtained for diamond tool burnishing:(6)$$Ra=2.117-0.345\times \frac{P-80}{50}+0.0375\times \frac{v-270}{90}-0.116118\times {\left(\frac{f-0.063}{0.031}\right)}^{2}\;-0.1611253\times {\left(\frac{P-80}{50}\right)}^{2}+0,045\times \frac{f-0.063}{0.031}\times \frac{P-80}{50}-0,37\times \frac{f-0.063}{0.031}\times \frac{v-270}{90}$$
has been verified based on the adequacy test of Fisher-Snedecor’s variance *F:*(7)$$F=\frac{S_{ad}^{2}}{S^{2}}$$
where $S^{2}$ is the variance of measurement errors, $S_{ad}^{2}$ is the adequacy variance calculated using the following formula:(8)$$S_{ad}^{2}=\frac{r\sum_{i\;=\;1}^{11}{\left(\bar{y_{i}}-\hat{y_{i}}\right)}^{2}}{n-k-1}$$
where *y_i_* is the mean value of the process coefficient in the *i*-th experiment, $\hat{y_{i}}$—value of the process coefficient calculated based on the regression equation for the levels of the input and output factors of the *i*-th experiment, *r* is the number of repetitions, *k* is the number of factors in the regression equation, and *n* is the number of experiments.

The coefficient *F* calculated from the Equation (7) was compared with the critical value *F_kr_* for the adopted level of significance α = 0.05:(9)$$F_{kr}=F_{\left(\alpha ;f_{1};f_{2}\right)}=F_{\left(0.05;7;11\right)}=3.0123$$

The values of the coefficients in the regression equation and their critical values are presented in Table 5. Finally, after simplification, the regression equation takes the form:(10)$$Ra=\;-0.42042-22.915\times f+0.00157\times P+0.00882\times v-120.83\times f^{2}-0.00006\times P^{2}-0.133\times f\times v+0.029\times P\times f$$

The results of Hartley’s plan show that the random factor *F* obtained did not exceed the critical value *F*_kr_ for the adopted level of significance α = 0.05. Therefore, the regression Equation (6) can be considered adequate. Hartley’s function reaches a minimum when *Ra* = 0.1042 μm (Figure 2), which corresponds to the experimental burnishing parameters ensuring the lowest value of the shaft surface roughness. In experiments no. 1–4, the relative error ranges from 18–24%, while in the remaining tests the error value does not exceed 3%.

The regression equation for SB with a cemented carbide tool is as follows:(11)$$\mathrm{Ra}=0.1296+0.527\times \frac{f-0.063}{0.031}-0.3387\times \frac{P-80}{50}+0.29725\times {\left(\frac{f-0.063}{0.031}\right)}^{2}+0.01712\times {\left(\frac{P-80}{50}\right)}^{2}-0.03025\times \frac{f-0.063}{0.031}\times \frac{v-270}{90}+0.05575\times \frac{P-80}{50}\times \frac{v-270}{90}$$

Equation (12), after simplification, takes the form:(12)$$Ra=\;0.3265192+0.73082\times f-0.005\times P-0.00032\times v+30.93\times f^{2}+0.000006848\times P^{2}-0.0108\times f\times v+0.0000012\times P\times v$$

The regression function 12 determined for the cemented carbide tool reaches a minimum of *Ra* = 0.063 μm for the following parameters: *f* = 0.032 mm/rev., *P* = 130 N, *v* = 180 rpm. (Table 6). In most cases, the values obtained experimentally differ only slightly from the values obtained in Hartley’s model, and the relative error is several percent. The highest error value equal to 33% was obtained for the set of burnishing parameters no. 3. Comparing the average values of the *Ra* obtained experimentally and using the regression Equation (12), it can be seen that for both samples the lowest values of *Ra* were achieved during experiment no. 3. The calculated factor *F* (Equation (7)) significantly exceeded *F_kr_* for the previously adopted significance level α = 0.05. Therefore, the Equation (12) for the cemented carbide tool cannot be considered adequate.

### 3.2. ANN Modelling

Due to the relatively small number of training sets for modelling the effect of burnishing parameters on the surface roughness of the shafts, networks with one hidden layer and a different number of neurons in this layer, i.e., 4, 8 and 13, were selected for the analysis of predictive possibility. It was initially assumed that if networks with these structures do not bring the expected results, their structure will be expanded. Too extensive a structure of the neural network to model a given problem may lead to its overfitting, and thus loss of the ability to generalize data. The analysis was based on data for a diamond tool whose Hartley’s model, determined by Equation (7), was statistically adequate.

During the training of neural networks, it was observed that the variable-metric algorithm provided the fastest convergence of the learning algorithm (Figure 3). The error level of 0.0005 with training network 3:3-13-1:1 was reached after 55 epochs, while the same error rate for the network with the smallest number of neurons in the hidden layer under consideration, i.e., 3:3-4-1:1 was reached after 159 learning epochs. The same error in learning the network with the Levenberg-Marquardt algorithm was obtained after 302, 206 and 82 learning epochs for networks 3:3-4-1:1, 3:3-8-1:1, 3:3-13-1:1 (Figure 4), respectively.

The number of learning epochs necessary to achieve the assumed error level for the network trained with the variable-metric and Levenberg-Marquardt algorithms is comparable, except that there are periodic fluctuations in the process of minimizing the network error by the LM learning algorithm (Figure 4). This is due to the fact that the LM algorithm works without having to compute the Hessian matrix [46]. The Hessian matrix is approximated by the Jacobian matrix containing first derivatives of the network errors with respect to the weights and biases. In these conditions the learning algorithm convergence depends on the vector of network errors [47]. The step size of the gradient descent in each epoch is selected automatically by the LM algorithm in order to achieve global convergence. With a small ratio of the number of pieces of training data to the number of neurons in the network, the LM algorithm is susceptible to reach a local minimum, however, due to the fact that the Hessian matrix is not computed, it can maintain directional stability.

Due to the very high tendency of the back propagation algorithm, it is recommended that the learning algorithm be stopped when the network error decreases no further [48]. Training of network 3:3-13-1:1 was stopped after 1180 epochs at the network error level of 0.0059 (Figure 5). The asymptotic nature of the course of changes in the value of the network training error proves that the minimum error value was reached. The training of the remaining networks was stopped at the same error value, although the course of the changes in network error during learning tends to further minimize the error. The increase in the number of neurons in the input layer, by introducing additional interneural connections, ensured faster convergence of the learning algorithms that were analysed. The observation of changes in the network learning error is a response to a possible network overtraining. Apart from the network training error, very important parameters indicating the network approximation abilities are the root mean square error, coefficient of determination and mean absolute error.

For all networks, the value of the coefficient of determination was at least *R*^2^ = 0.7505 accurate to three significant figures (Table 7). The lowest value of the *RMSE* and *MAE* for all the networks examined was assured by the qN algorithm. It should be considered that the best network for modelling the effect of slide burnishing parameters on the average surface roughness value of the treated surface is the multilayer perceptron 3:3-8-1:1 trained with the variable-metric algorithm, due to the high value of the determination coefficient *R*^2^, and simultaneously, to the lowest of both the *RMSE* and *MAE* errors for the training set.

The regression response of network 3:3-8-1:1 is presented in Table 8. The value of the network error does not exceed 0.0022 μm in any of the experiments conducted. Thus, taking into account the error values resulting from the response of Hartley’s model shown in Figure 4, neural modelling is a more effective alternative. Despite the small number of training sets, the neural model trained with different methods provided a better prediction of the arithmetic mean value of the profile ordinates than Hartley’s model. It suggests that a strong dependence exists between SB parameters and the shaft surface. So, the experimental training data were not strongly noised.

The increase in the pressure of the burnishing tool, under the same feed values, causes a decrease in the surface roughness of the shafts (Figure 6a). This conclusion is in agreement with the results of Dzierwa and Markopoulos [8] who machined 42CrMo4 steel surfaces and Shiou et al. [49] who processed SUS420J2 stainless steel shafts. Plastic deformation is caused by the forces causing surface pressures exceeding the value of the yield stress of the processed material [10]. Pressure exerted through a very hard and very smooth roller or ball on a surface generates a flattening of surface asperities and work hardening of the buried impurity layer [14]. As a result of this process, the surface roughness decreases, and the surface layer strengthens [4,18,50]. The mean values of the burnishing pressure and speed can be employed to enhance the quality of machining [51]. The effect of displacing the surface asperities is a reduction in the roughness of the treated surface. After the prior turning operation, deformations produced by sliding on the peak zone cause the valleys to remain unfilled, and for traces to remain on the surface in the form of recesses [52]. During burnishing, the soft material is not only deformed on the surface, but also to a significant depth in the surface layer. Too high a value can lead to peeling, which is accompanied by a rapid increase in roughness [53]. Too much pressure of the burnishing tool can worsen the quality of the treated surface due to the possibility of the appearance of surface defects and the presence of stress corrosion [53].

In general, reducing the feed rate at low burnishing speed decreases the *Ra* parameter of the shaft surface (Figure 6b). Reducing the impact force of the burnishing tool at a constant value of burnishing speed ensures a reduced roughness of the shaft surface (Figure 6c). At small values of the impact force of the burnishing tool, the increase in the burnishing tool led to a slight increase in the surface roughness parameter *Ra*. At high values of impact force there is the opposite relationship between burnishing speed and surface roughness. However, there are complex interactions between feed rate and burnishing speed. Higher values of the *Ra* parameter occur with a combination of high feed rate and low burnishing speed. The increase of the feed rate in the range of 180–200 mm/min while maintaining a constant burnishing speed initially led to an increase in the distance between the burnishing passes and therefore surface roughness increases. A further increase in burnishing speed causes a closeness of adjacent burnishing passes and, as a result, average surface roughness *Ra* decreases. This phenomenon was also observed by Ibrahim [51] and Dwivedi et al. [54]. Higher roller burnishing speed increases the surface temperature of a workpiece [54,55] and the material forming the asperities is more susceptible to plastic deformation.

One of the main limitations of the neural network model used is the ability to predict the value of the average surface roughness of the treated surface for the range of values of SB parameters used in the network training process. In this paper only one parameter (*Ra*) in the network output was considered due to the limited amount of training data. It is possible to include many parameters in the output of the network. Any measurable parameters may be included in the network. However, with the addition of a new parameter for network input and/or output, the data size requirements for the training set increase exponentially.

## 4. Conclusions

The article describes the effect of the parameters of the slide burnishing of 42CrMo4 steel shafts on their surface roughness using both Hartley’s plan and artificial neural networks. Although the training set contained only 11 measurement data, neural networks trained with three different algorithms provided the possibility of predicting the value of the average surface roughness of the shaft profile at the *R*^2^ level of at least 0.7505. The ANN with the best regression statistics predicted the average surface roughness of shafts with *R*^2^ = 0.987. The variable-metric algorithm ensured a high efficiency of the training process determined by the network error. The lowest values of *RMSE* and *MAE* errors were obtained in all of the network structures analysed that were trained with the quasi-Newton algorithm: 3:3-4-1:1, 3:3-8-1:1, 3:3-13-1:1. The regression statistics and the errors in predicting the average surface roughness of the shafts with 3:3-8-1:1 are much better than the results obtained with Hartley’s plan.

The neural model clearly shows the dependence in which an increase in the impact force results in a decrease of the surface roughness of the shaft surface. The coeffect of burnishing speed and feed rate is more complex and is represented by a “saddle” response surface. Low feed rate with low burnishing speed leads to a small value of the *Ra* parameter. However high values of the feed rate with small values of the burnishing speed produces the highest values of surface roughness.

In future research, slide burnishing of the shafts should be carried out with a wide range of changes in machining parameters. The next task will be to take into account the influence of tool roughness and lubrication conditions on the surface roughness of the shafts at the input of the neural network. By increasing the range of values of the input parameter and the number of training sets, it will be possible to gain network forecasting ability beyond a certain range of data used in the training process.

## Figures and Tables

**Figure 1 materials-14-01175-f001:**
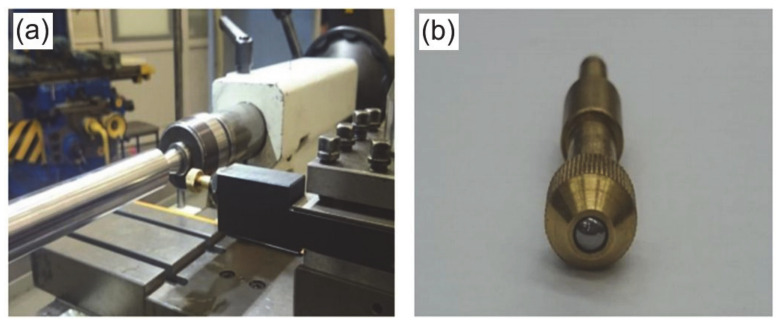
View of the test stand (**a**) and burnishing tool (**b**).

**Figure 2 materials-14-01175-f002:**
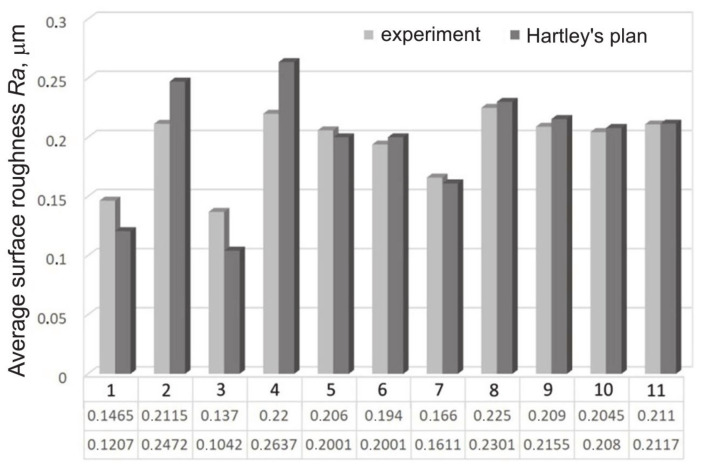
Comparison of the experimental value of the average surface roughness *Ra* of the shafts with the value resulting from Hartley’s plan.

**Figure 3 materials-14-01175-f003:**
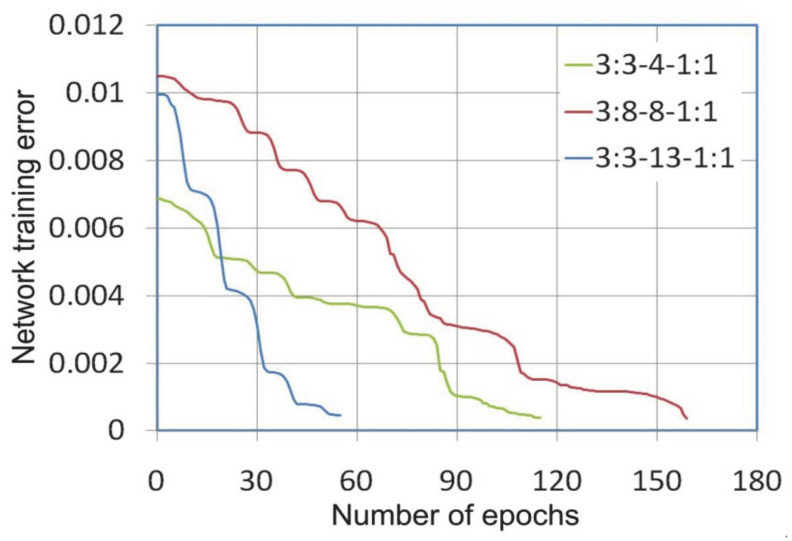
The network training error vs. number of epochs during the learning process with the variable-metric algorithm.

**Figure 4 materials-14-01175-f004:**
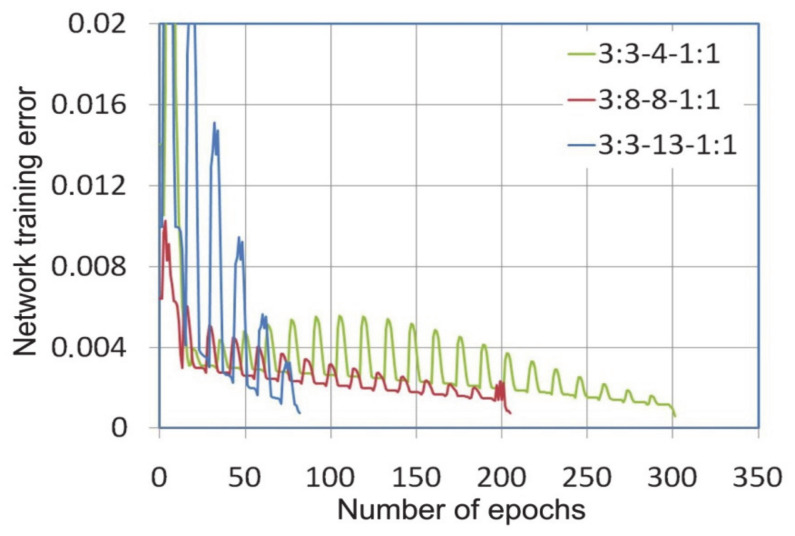
The network training error vs. number of epochs during the learning process with the Levenberga-Marquardt algorithm.

**Figure 5 materials-14-01175-f005:**
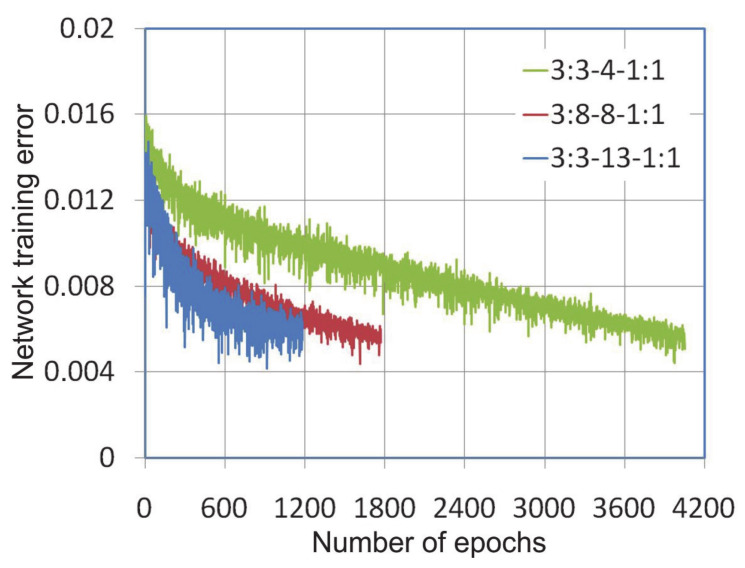
The network training error vs. number of epochs of the learning process with the BP algorithm.

**Figure 6 materials-14-01175-f006:**
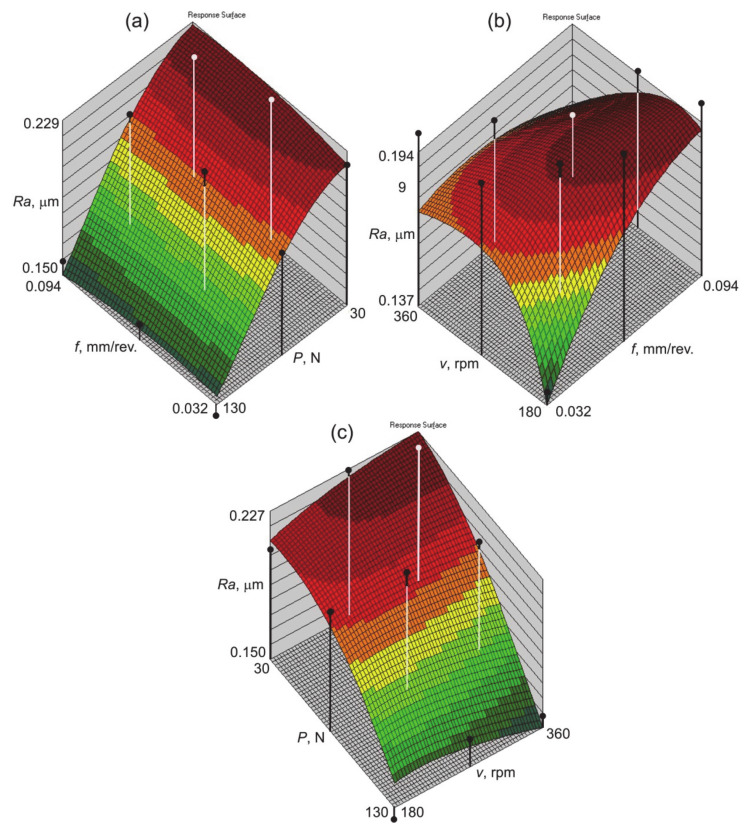
Response surfaces of the neural network 3:3-8-1:1 trained with the qN algorithm showing the influence of (**a**) feed rate and impact force, (**b**) feed rate and burnishing speed, and (**c**) impact force and burnishing speed on the average surface roughness *Ra.*

**Table 1 materials-14-01175-t001:** Basic mechanical properties of 42CrMo4 steel.

Yield Stress R_p0.2_, MPa	Ultimate Tensile Stress R_m_, MPa	Elongation, %	Hardness HB	Toughness KV, J
650	900–1100	12	265–325	min. 35 at 20 °C

**Table 2 materials-14-01175-t002:** Matrix plan of experiments.

Number of Experiment	Feed Rate *f*, mm/rev.	Impact Force *P*, N	Burnishing Speed *v*, rpm
1	0.094	130	360
2	0.094	30	180
3	0.032	130	180
4	0.032	30	360
5	0.094	80	270
6	0.032	80	270
7	0.063	130	270
8	0.063	30	270
9	0.063	80	360
10	0.063	80	180
11	0.063	80	270

**Table 3 materials-14-01175-t003:** Average surface roughness *Ra* of shafts burnished using a diamond tool.

Experiment No.	Average Surface Roughness *Ra*			Error Variance $S^{2}$
	Measurement 1	Measurement 2	Mean	
1	0.145	0.148	0.1465	0.000045
2	0.216	0.207	0.2115	0.000405
3	0.143	0.131	0.137	0.00072
4	0.217	0.223	0.220	0.00018
5	0.205	0.207	0.206	0.00002
6	0.191	0.197	0.194	0.00018
7	0.169	0.163	0.166	0.00018
8	0.226	0.224	0.225	0.00002
9	0.216	0.202	0.209	0.00098
10	0.204	0.205	0.2045	0.000005
11	0.208	0.214	0.211	0.00018

**Table 4 materials-14-01175-t004:** Average surface roughness *Ra* of shafts burnished using a cemented carbide tool.

Experiment No.	Average Surface Roughness *Ra*			Error Variance $S^{2}$
	Measurement 1	Measurement 2	Mean	
1	0.209	0.202	0.2055	0.00002
2	0.271	0.258	0.2645	0.00008
3	0.0936	0.0914	0.0925	0.000002
4	0.156	0.153	0.1545	0.000005
5	0.217	0.190	0.2035	0.0004
6	0.107	0.114	0.1105	0.00002
7	0.110	0.0966	0.1033	0.00009
8	0.178	0.193	0.1855	0.0001
9	0.125	0.120	0.1225	0.00001
10	0.130	0.142	0.136	0.00007
11	0.133	0.143	0.138	0.00005

**Table 5 materials-14-01175-t005:** The significance of parameters in the regression equation describing SB with a diamond tool.

Coefficient	|Value|	Relationship	Critical Value	Significance
*b_o_*	|2.117|	>	0.0442	significant
*b_1_*	|0.0217|	<	0.03271	not significant
*b_2_*	|−0.345|	>	0.03271	significant
*b_3_*	|0.0375|	>	0.03271	significant
*b_11_*	|−0.116118|	>	0.051492	significant
*b_22_*	|−0.1611253|	>	0.051492	significant
*b_33_*	|−0.04862|	<	0.051492	not significant
*b_12_*	|0.045|	>	0.04006	significant
*b_13_*	|−0.37|	>	0.04006	significant
*b_23_*	|0.0025|	<	0.04006	not significant

**Table 6 materials-14-01175-t006:** The significance of parameters in the regression equation describing SB with a cemented carbide tool.

Coefficient	|Value|	Relationship	Critical Value	Significance
*b_o_*	|0.1296|	>	0.007454849	significant
*b_1_*	|0.0527|	>	0.0527	significant
*b_2_*	|−0.03387|	>	0.0527	significant
*b_3_*	|−0.00175|	<	0.0527	not significant
*b_11_*	|0.029725|	>	0.008684493	significant
*b_22_*	|0.01712|	>	0.008684493	significant
*b_33_*	|0.00196901|	<	0.008684493	not significant
*b_12_*	|0.00075|	<	0.006756254	not significant
*b_13_*	|−0.03025|	>	0.006756254	significant
*b_23_*	|0.05575|	>	0.006756254	significant

**Table 7 materials-14-01175-t007:** Parameters of the training process of the networks examined.

Network	Training Algorithm	Training Set		Verification Set		R2
		RMSE	MAE	RMSE	MAE	
3:3-4-1:1	BP	0.02316	0.01388	0.04760	0.03942	0.7896
	qN	0.02091	0.02091	0.00737	0.00607	0.8704
	LM	0.02682	0.01827	0.01007	0.00910	0.7898
3:3-8-1:1	BP	0.01456	0.01259	0.01446	0.01313	0.9661
	qN	0.00305	0.00250	0.00303	0.00260	0.9971
	LM	0.00607	0.00505	0.00357	0.00254	0.9876
3:3-13-1:1	BP	0.02855	0.01966	0.00247	0.00228	0.8946
	qN	0.02612	0.01855	0.00200	0.00189	0.8826
	LM	0.02806	0.01992	0.00316	0.00282	0.7505

**Table 8 materials-14-01175-t008:** Comparison of the average surface roughness *Ra* obtained experimentally with the results of the 3:3-8-1:1 neural network model trained with the variable-metric algorithm (results for experiments included in the training set).

Experiment No.	Average Surface Roughness *Ra*, μm		Error
	Experiment	ANN 3:3-8-1:1	
1	0.1465	0.168572	0.022072
3	0.137	0.159752	0.022752
4	0.22	0.233345	0.013345
5	0.206	0.218104	0.012104
6	0.194	0.213618	0.019618
7	0.166	0.165331	−0.00067
9	0.209	0.220224	0.011224
10	0.2045	0.210982	0.006482
11	0.211	0.216063	0.005063

## Data Availability

The data presented in this study are available on request from the corresponding author.

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
