# Peer review of "Modelling the Influence of Slide Burnishing Parameters on the Surface Roughness of Shafts Made of 42CrMo4 Heat-Treatable Steel"

_materials, 2021, doi:10.3390/ma14051175_

Round 1

Reviewer 1 Report

The paper proposes the modelling of the Slide Burnishing (SB) by comparing Hartley's plan and the artificial neural networks (ANN), and finding better results for this latter modelling. The paper is quite interesting, being the ANN methods deeply investigated, and proposing an adequate set of experimental data. Some issues and cues by the Reviewer are suggested below:
- The Abstract is too long, 290 words instead of 200 (max).
- In the Abstract it is recommended to avoid special symbols and not defined acronyms.
- In the Introduction the details of SB are well described. However, it should be interesting to provide some comparisons with similar techniques, such as the so called LPB (Low Plasticity Burnishing) in which the ball at the tool tip freely rotates by the effect of a pressurized fluid, so drastically reducing the friction.
- In the Experimental section, the 42CrMo4 steel is introduced and it is mentioned the expression 'high ductility of the core'. The quenched and tempered steel are used for having a good balance between toughness and strength, but not case hardened. Thus the surface and the core properties are the same.
- Eqs. 7 and 10 are presented in different forms, and in the end Eq. 10 is said to be not adequate. How the proposed form was obtained? and could the form of Eq. 7 be effective also for the carbide tool data?
- Were the fluctuations of Fig. 6 expected and could be explained in some sort by considering the Levenberga-Marquardt algorithm?
- In the Figs. 8 it is recommended to show the experimental values as dots superimposed to the prediction surfaces for comparison.
- [general comment 1] All the experimental data was used as training in the ANN algorithm. Was attempted or considered to apply the ANN just to a subset of the experimental data, and then use the remaining tests for a more independent verification?
- [general comment 2] In this paper the R_a is only considered, while other surface properties are just mentioned but not used, such as the microhardness. Could it be possible to apply these methodologies also to the microhardness surface property?

Author Response

Dear Reviewer

We would like to thank the Reviewers for careful and thorough reading of this manuscript and for the thoughtful comments and constructive suggestions, which helped us to improve the quality of this manuscript. We have carefully checked  and addressed all the comments as explained below.

The changes in the manuscript are highlighted in yellow.

Regards,

The authors

Reviewer 2 Report

Title: Modeling the Influence of Slide Burnishing Parameters on the Surface Roughness of Shafts Made of 42CrMo4 Heat-treatable Steel Using Artificial Neural Networks

Manuscript #: 1110977

In the paper " Modeling the Influence of Slide Burnishing Parameters on the Surface Roughness of Shafts Made of 42CrMo4 Heat-treatable Steel Using Artificial Neural Networks ", the authors investigated the influence of the SB process on the surface topography of 42CrMo4 steel shafts on the basis of Hartley’s PS/DS-P:Ha3 plan. Artificial neutral network models were also applied to predict the roughness of the surface of the shafts after SB. The influence of the input parameters such as pressure, burnishing speed and feed rate are taken into consideration. Some interesting conclusions are drawn. Before this manuscript can be taken into further consideration, the authors should respond to the following comments/suggestions.

  • In the abstract, SB should be defined before being used.
  • In the Introduction section, the authors listed a lot of studies on the topic in this paper, while there are few discussions about the limitations for the current investigation methods and the merits of the ANN method.
  • The application of the influence of SB parameters on the surface roughness of shafts made of 42CrMo4 steel should be described in the Introduction part.
  • In table 2, what do you mean by “Diameter D”? Is diameter a basic mechanical property?
  • Section 2 and 3 should be combined. Some information can be moved to Introduction part. Meanwhile, a lot of descriptions are tedious, especially for Section 3, we can find better information in a textbook. The authors should condense this part.
  • Section 4 should be “Results and Discussions”
  • The title, which only emphasizes the ANN model, is not quite suitable for the content in this paper.
  • For the ANN, what would happen if the number of training sets is increased?

Author Response

Dear Reviewer

We would like to thank the Reviewers for careful and thorough reading of this manuscript and for the thoughtful comments and constructive suggestions, which helped us to improve the quality of this manuscript. We have carefully checked  and addressed all the comments as explained below.

The changes in the manuscript are highlighted in blue.

Regards,

The authors

Reviewer 3 Report

This paper entitled “Modelling the Influence of Slide Burnishing Parameters on the Surface Roughness of Shafts Made of 42CrMo4 Heat-treatable Steel Using Artificial Neural Networks", the authors study the slice burnishing process on the surface topography of 42CrMo4 steel shafts and correlation between the processing parameters and the surface roughness of the specimens. The topic is of interest to the Material’s due to describing the effect of the parameters of the slide burnishing of 42CrMo4 346 steel shafts on their surface roughness. However, I have to recommendations and suggestion to strengthen this work before consider it for publication:

  1. Introduction section. The intro is clear with respect to the studies performed by others authors and the one presented here, although I miss some recent articles that have explained how this burnishing process is important for the industry and particular applications that will improve surface components or can be associated with other manufacturing process to enhance other the surface properties or residual stresses. See these two articles: Doi:10.1016/j.surfcoat.2019.04.010 and Doi:10.1016/j.jmst.2017.11.041.

  1. The weakness part of this article is that there is not discussion, accordingly it is necessary to compare the obtained results with respect to the prior articles. In this sense, it will be able to spotlight the importance of the proposed model and the range of the manufacturing parameters in presented in this work. On top of that, it is necessary to describe the limitation of the ANN used to correlate the surface topography based on the used operating parameters.  

Author Response

Dear Reviewer

We would like to thank the Reviewers for careful and thorough reading of this manuscript and for the thoughtful comments and constructive suggestions, which helped us to improve the quality of this manuscript. We have carefully checked  and addressed all the comments as explained below.

The changes in the manuscript are highlighted in green.

Regards,

The authors

Round 2

Reviewer 2 Report

The reviewer is satisfied with the revised manuscript, which should be acceptable by "Materials"

Reviewer 3 Report

The revised manuscript have resolved all the initial concerns/suggestions, consequently, this paper can be accepted in its present form.